# SLIM OBJECT DIRECT OFFSET (SODO): WHY YOLO WITH TOOD STRUGGLES TO DETECT SLIM OBJECTS? A NEW MATCHING APPROACH FOR SLIM OBJECTS

## ABSTRACT

The YOLO model has become the mainstream in object detection. The latest YOLOv8 to YOLOv13 models all use TOOD as the target matching mechanism and have achieved excellent performance in object detection tasks. However, certain limitations remain. Specifically, when detecting slim objects (bounding box (bbox) aspect ratio > 20), it is difficult for TOOD to match the grids. This is due to TOOD's matching mechanism. According to TOOD matching mechanism, for slim objects, they are likely to not match any grid in the large or medium detector, as slim bboxes probably do not pass any grid center. To solve this problem, we propose the Slim Object Direct Offset (SODO). By designing strategies like Slim Object Assignment (SOA), Direct Offset, Center Offset, and Layer Wise Assignment (LWA), slim objects can match grids in the large or medium detector. Our SODO method outperforms the baseline model (YOLOv11-L) on both COCO (+0.3% mAP@50) and COCOSlim (+1% mAP@50) datasets. Additionally, for slim objects, significant performance improvements are achieved on our synthetic slim object dataset and multiple public datasets from Roboflow, including pole_detection and tans.

## 1 INTRODUCTION

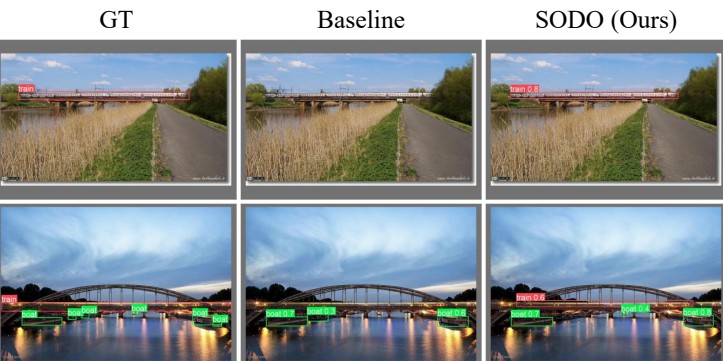

Figure 1: **Results examples of baseline and our model.** YOLO model based on the TOOD (YOLOv11-L) has poor detection performance for slim objects, while our SODO can detect slim objects effectively.

Real-time object detection is a critical task in computer vision (Cheng et al., 2023; Zou et al., 2023), aiming to localize and classify objects given images with minimal latency—a critical capability for applications ranging from industrial anomaly detection to autonomous driving and medical image processing (Vijayakumar & Vairavasundaram, 2024). Recent advances have solidified the dominance of single-stage convolutional neural network (CNN) detectors that unify region proposal, classification, and regression into end-to-end frameworks. (Girshick et al., 2014; He et al., 2017;

Redmon et al., 2016; Zhao et al., 2024). Among them, YOLO (You Only Look Once) series (Redmon et al., 2016; Wang et al., 2023) has become mainstream due to its excellent balance between inference speed and accuracy. Specifically, recent YOLO series (YOLOv8-YOLOv13) (Varghese & M., 2024; Wang et al., 2024b;a; Khanam & Hussain, 2024; Tian et al., 2025; Lei et al., 2025), which utilize TOOD (Feng et al., 2021) as the object matching mechanism, have demonstrated outstanding detection performance in the MS COCO datasetLin et al. (2014). However, in practice, it has been observed that YOLO models based on TOOD exhibit subpar performance in detecting slim objects.

Although some image datasets such as MS COCO only have a small proportion of slim objects, datasets in industrial inspection, autonomous driving, and other specialized fields may contain a lot of slim objects. Therefore, addressing the detection of slim objects is of substantial importance.

As depicted in Figure 1, baseline performs poorly in detecting slim objects. This baseline is YOLOv11-L model, unless otherwise specified, the baseline model refers to YOLOv11-L. We analyze the YOLO framework (as shown in Figure 9 in the Appendix) and the TOOD mechanism. Through our analysis, we find that TOOD probably fails to match the bbox to any grid in the large and medium detectors. Consequently, these slim objects suffer from insufficient learning during the training process, resulting in poor detection results for slim objects. We propose a method called Slim Object Direct Offset (SODO) to solve the problem. Our contributions are summarized as follows:

1. We propose the Slim Object Assignment (SOA) to ensure that slim objects can be matched to the grids in the deeper detectors with large receptive fields during the training phase.

2. To adapt to the SOA, we proposed two new ways to represent predicted bbox (pred bbox): 1. Direct Offset and 2. Center Offset.

3. We proposed Layer Wise Assignment (LWA) to normalize the target scores of three detectors independently. This allows the deeper detectors can catch up with the shallow detectors in the early training stage. After training for a certain period, we close this LWA so that each detector can be responsible for predicting objects of different sizes.

4. It is beneficial to remove overlaps from the final target score to facilitate the model's rapid acquisition of higher predicted scores during the initial stages of training.

5. We constructed a synthetic dataset of slim objects, which is helpful to analyze and evaluate the detection performance on slim objects.

## 2 METHOD

To address TOOD matching problem for slim objects, we propose a new matching method called Slim Object Direct Offset (SODO). This approach contains three key strategies: SOA, Direct Offset, Center Offset, and LWA.

### 2.1 SLIM OBJECT ASSIGNMENT (SOA)

TOOD's matching mechanism requires that the target bbox must pass through the center of a grid to match the current grid. For common cases, targets with normal aspect ratio can be matched to grids based on this mechanism. However, when the target has a slim shape, it's likely that it cannot match any grid, which depends on the position of the slim target, as is shown in Figure 2(a). In addition, detecting slim targets requires a large receptive field, so even if it can be matched in the small detector, it is difficult to be fully learned. To address this problem, we propose the SOA matching mechanism. As long as the target bbox passes through a grid, the target bbox matches the grid, as is shown in Figure 2(b). By this method, it can ensure that slim targets can be matched to grids in large, medium, and small detectors. The difference between TOOD's and our method's matching mechanism is shown in Figure 2(c), represented by Equation 1 and Equation 2 respectively.

$$grid_{TOOD} = \begin{cases} Pos, & \mathbb{I}(x_c \geq x_{lt}^t) + \mathbb{I}(x_{rd}^t \geq x_c) + \mathbb{I}(y_c \geq y_{lt}^t) + \mathbb{I}(y_{rd}^t \geq y_c) = 4 \\ Neg, & \mathbb{I}(x_c \geq x_{lt}^t) + \mathbb{I}(x_{rd}^t \geq x_c) + \mathbb{I}(y_c \geq y_{lt}^t) + \mathbb{I}(y_{rd}^t \geq y_c) < 4 \end{cases} \quad (1)$$

$$grid_{Ours} = \begin{cases} Pos, & \mathbb{I}(x_{rb}^t \geq x_{lt}^g) + \mathbb{I}(x_{rb}^g \geq x_{lt}^t) + \mathbb{I}(y_{rb}^t \geq y_{lt}^g) + \mathbb{I}(y_{rb}^g \geq y_{lt}^t) = 4 \\ Neg, & \mathbb{I}(x_{rb}^t \geq x_{lt}^g) + \mathbb{I}(x_{rb}^g \geq x_{lt}^t) + \mathbb{I}(y_{rb}^t \geq y_{lt}^g) + \mathbb{I}(y_{rb}^g \geq y_{lt}^t) < 4 \end{cases} \quad (2)$$

where $bbox_{GT} = (x_{lt}^t, y_{lt}^t), (x_{rb}^t, y_{rb}^t)$ is the GT bbox with top left and bottom right coordinates, $bbox_{Grid} = (x_{lt}^g, y_{lt}^g), (x_{rb}^g, y_{rb}^g)$ is the grid bbox with top left and bottom right coordinates. $Center_{Grid} = (x_c, y_c)$ is the center point of the Grid, $\mathbb{I}(*)$ is the indicator function. Its value is 1 when the conditions are true; otherwise, it is 0.

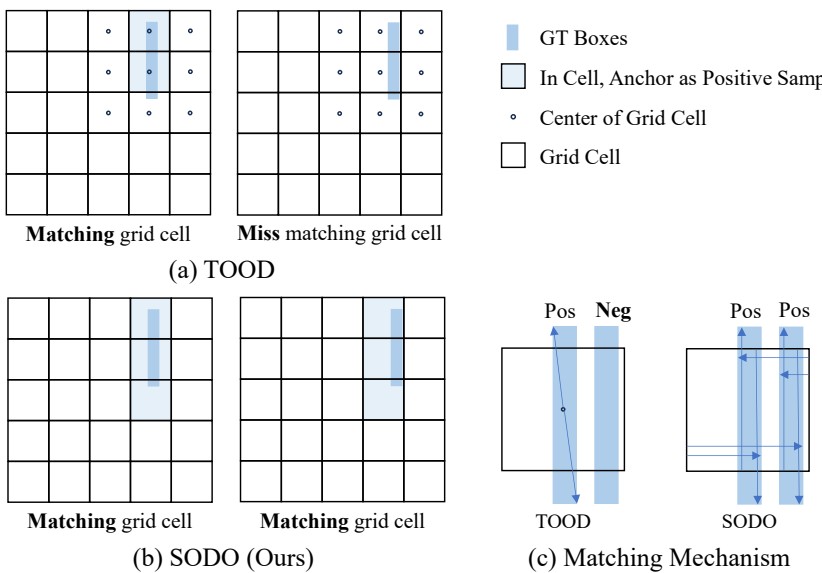

Figure 2: TOOD and SODO (Ours).

## 2.2 NEW REPRESENTATION FOR BBOX

To cooperate with SOA method, we need to build a new representation for the pred bboxes: since the original TOOD method only matches the bboxes that pass through the center of the grid, the pred bboxes can be represented by the top-left (tl) & right-bottom (rb) offset to the grid center, and these offsets' x and y are non-negative values. But our new matching method does not ensure that the bboxes pass through the grid center, so the offset's x and y may be negative values. We propose two ways to represent the pred bboxes.

### 2.2.1 DIRECT OFFSET

The recent YOLO models use Distribution Focal Loss (DFL) (Li et al., 2023) to represent tl & rb offsets, that is, it uses a distribution to represent these offsets instead of a definite value, as is shown in Equation 3. For more details about DFL, please refer to A.3 in Appendix. By modeling the position of the bbox as a distribution, it enhances the accuracy of bbox regression through learning this distribution. We can see that the tl and rb offset values in the DFL are all non-negative. While in our SODO method, the tl and rb offsets can be negative, as is shown in Figure 3(a). Notice that these negative values will not be lower than -0.5 (can not reach the adajcent grid center), we can simply subtract 1 from regmax, as is shown in Equation 4.

$$\Delta_t = \sum_{i=0}^{regmax} i \times p_i^t, \Delta_l = \sum_{i=0}^{regmax} i \times p_i^l,$$

$$\Delta_b = \sum_{i=0}^{regmax} i \times p_i^b, \Delta_r = \sum_{i=0}^{regmax} i \times p_i^r \quad (3)$$

$$\Delta_t = \sum_{i=-1}^{regmax-1} i \times p_i^t, \Delta_l = \sum_{i=-1}^{regmax-1} i \times p_i^l,$$

$$\Delta_b = \sum_{i=-1}^{regmax-1} i \times p_i^b, \Delta_r = \sum_{i=-1}^{regmax-1} i \times p_i^r \tag{4}$$

Where $regmax$ denotes the number of discrete probability value, $p_i^*$ denotes the i-th probability for direction *, $\sum_i p_i^* = 1$. $\Delta$ denotes the offset, t, l, b, r denote top, left, bottom, right respectively. The notations are the same in Equation 3 and Equation 4.

Notice that this method cannot guarantee the lt coordinate must be in the upper left of the rb coordinate, especially when the pred bbox is a slim bbox, but we never find this problem in our experiments.

### 2.2.2 CENTER OFFSET

This method aims to ensure that the offsets remain non-negative. To achieve this, we introduce an additional center offset to represent the offset between the pred bbox and the grid center. Figure 3 shows a comparison between Direct Offset and Center Offset.

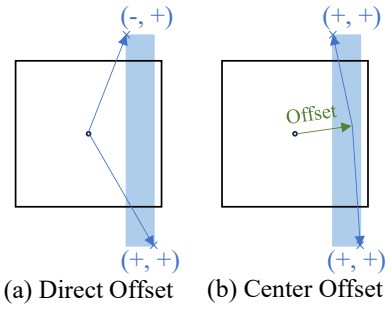

(a) Direct Offset  (b) Center Offset

Figure 3: Direct Offset and Center Offset.

The Center Offset representation of pred bbox is shown in Equation 5.

$$\begin{cases} x'_{cg} = x_{cg} + \Delta x \\ y'_{cg} = y_{cg} + \Delta y \end{cases} \tag{5}$$

Where $x_{cg}$ and $y_{cg}$ are respectively the x and y of the grid center coordinate, $\Delta x$ and $\Delta y$ are respectively the center offset of $x$ and $y$, $x'_{cg}$ and $y'_{cg}$ are respectively the $x$ and $y$ of the new grid center coordinate.

We tried these two pred bbox presentations and found that the convergence speed of the center offset was much slower. It might be because it requires additional learning of the center offsets compared to the baseline and Direct Offset. Therefore, the task becomes more difficult and converges more slowly.

Furthermore, we found that if our method uses mix precision training, NAN loss is more likely to occur than baseline. We located and found that it occurs when calculating the $(x - \mu)/\sigma$ of Batch Normalization (BN) (Ioffe & Szegedy, 2015). The value of x is very large, but the values of $\mu$ and $\sigma$ are very small. The values after normalization are not within -1 and 1, but instead are very large values. They exceed the range represented by float16, which results in INF value, and then INF - INF leads to NAN. Due to the transitive nature of NAN, it eventually leads to loss of NAN. For now, we have not been able to solve this problem elegantly. We just turned off the mix precision training to work around this problem.

## 2.3 LAYER WISE ASSIGNMENT (LWA)

Notice that for the slim target here, it requires a larger receptive field. Therefore, it is more appropriate to be detected in the large or medium detectors. In TOOD, it defines align metric to build the target score, instead of a hard label as is shown in Equation 6. For more details about align metric, refer to sectionA.2.2 in Appendix. The target score will be normalized among the grids of the large, medium, and small detectors together. Notice that the information paths to the large and medium detectors are longer than to the small detectors, making them more difficult to learn at the beginning of training. Therefore, the predicted score (pred score) of the large and medium detectors at early training stage may be much smaller than that of the small detector. According to the representation of the target score in Equation7, this will lead to that the target score of small detector has a dominant advantage over the medium and large detectors, which in turn suppress the learning of medium and large detector. Therefore, we propose LWA which performs normalization independently for small, medium, and large detectors at the beginning of training, as is shown in Equation8.

$$t = s^\alpha \times u^\beta \tag{6}$$

$$t_{norm} = \frac{t}{max_{s,m,l}(t)} \tag{7}$$

where $s$ and $u$ denote a classification score and an IoU value, respectively. $\alpha$ and $\beta$ are used to control the impact of the two tasks in the anchor alignment metric. s, m, l denote the small, medium, large detectors' target score respectively. $max_*(t)$ denotes the maximum target score of t among detector *. $t_{norm}$ denotes the normalized target score. Notations' meaning are all the same for all the Equations.

$$t_{s\_norm} = \frac{t}{max_s(t)}$$

$$t_{m\_norm} = \frac{t}{max_m(t)} \tag{8}$$

$$t_{l\_norm} = \frac{t}{max_l(t)}$$

Consider that if LWA is applied through the whole training phase, it is possible that the shallow detector may also detect some low-quality bboxes. For instance, the small detector may also detect long bboxes, but they're far less accurate. Therefore, we only apply LWA at the early stage of training, we found that it achieves the best performance when only applying the LWA in the first 20% epochs.

## 2.4 REMOVE OVERLAPS

We found that in the YOLO code with TOOD, after obtaining the normalized target score, this target score then mutiplies with overlaps again to obtain the final target score as is shown in Equation 9.

$$t_{final} = t_{norm} \times u \tag{9}$$

This will result in the maximum value of the target score being u instead of 1, where u is the overlap between target bbox and pred bbox, which is very small at early training stage, that is, the maximum target score will be very low at early training stage, which will lead to hard learning for pred score. We found that after removing overlaps, $mAP$ improves.

## 3 EXPERIMENT AND ANALYSIS

### 3.1 DATASETS

To verify that our method is superior to the baseline model, we do evaluation on (1) COCO2017, (2) COCOSlim filtered out from the COCO dataset with an aspect ratio greater than 20, (3) public datasets obtained from the Roboflow: pole_detection (Poleproject, 2024) and tans (2Crack500, 2025) (4) we build synthetic datasets: slim objects and normal objects based on GitHub code (QuantumForgeEngineer, 2024-04-17). Among them, slim objects are slim rectangle with aspect ratio around 80. Normal objects are rectangles with aspect ratio is around 1:1 and circles of different sizes. Unlike QuantumForgeEngineer (2024-04-17) constructing synthetic dataset for OBB tasks, what we

construct is undirected object detection data. For the convenience of analysis, we only generate a single one object for each synthetic image. Except the AMP setting(we set to False as mentioned previously), we use exactly the same hyperparameter settings as the original YOLO models.

## 3.2 ABLATION EXPERIMENT

We evaluate the quantification precision of the following different strategy combinations as Table 1. It can be seen that Direct Offset equiped with 20%time LWA achieves the best performance. Here we can see that the effect of Strategy 2 is much worse than that of Strategy 3. This might be because the Center Offset method adds extra offset, making the learning task much more difficult and requiring longer epochs for convergence. In addition, the mAP of Strategy 4 is the worst. This is mainly because its detection results contain a large number of low-quality bbox by small detector, resulting in a very low precision and subsequently a very low mAP.

Table 1: Ablation experiments in slim object dataset

|  | Remove overlaps | SOA | New representation | LWA | mAP@50 | mAP@50-95 |
|---|---|---|---|---|---|---|
| Baseline | - | - | - | - | 0.939 | 0.513 |
| Strategy 1 | ✓ | - | - | - | 0.944 | 0.516 |
| Strategy 2 | ✓ | ✓ | ✓(Center Offset) | - | 0.948 | 0.540 |
| Strategy 3 | ✓ | ✓ | ✓(Direct Offset) | - | 0.995 | 0.864 |
| Strategy 4 | ✓ | ✓ | ✓(Direct Offset) | ✓(Full time) | 0.773 | 0.550 |
| Strategy 5 | ✓ | ✓ | ✓(Direct Offset) | ✓(Close at 50% epoch) | 0.995 | 0.872 |
| Strategy 6 | ✓ | ✓ | ✓(Direct Offset) | ✓(Close at 20% epoch) | **0.995** | **0.887** |

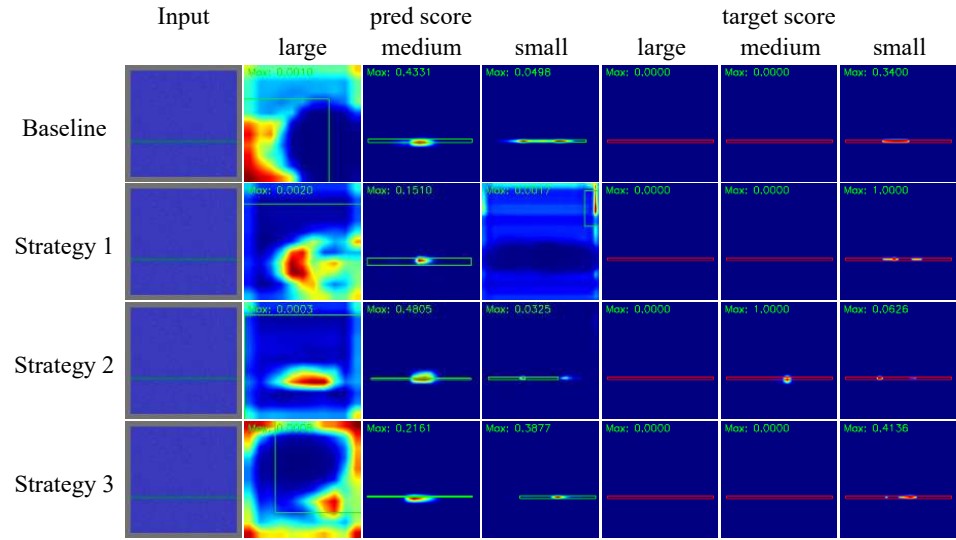

Figure 4: Visualization of detectors' outputs at the 10th epoch in slim object dataset. rmo is abbreviated for remove overlaps

To get a better insight of the strategies we proposed, we visualize the target score and pred score output by the detectors. In Figure 4, from top to bottom, they correspond to Baseline, Strategy 1, Strategy 2, Strategy 3 in Table 1, from left to right, they are input and *large detector pred score, medium detector pred score, small detector pred score, large detector target score, medium detector target score, small detector target score*. The upper left corner of each heatmap shows the maximum value in the heatmap. The box in the pred score heatmap is the pred bbox at the position where the maximum value is located and the box in the target score heatmap is the ground truth bbox.

Comparing the target scores of the small detectors of Strategy 1 and Baseline in Figures 4, it can be seen that after removing overlaps here, the target score increases significantly. Furthermore, we can see that the target score of medium detector of Strategy 2 is 1 rather than 0, indicating that it

matches grid in medium detector. Since overlaps between pred bbox and target bbox may be very small at early training stage, the target score of some detectors is very low, or even 0.

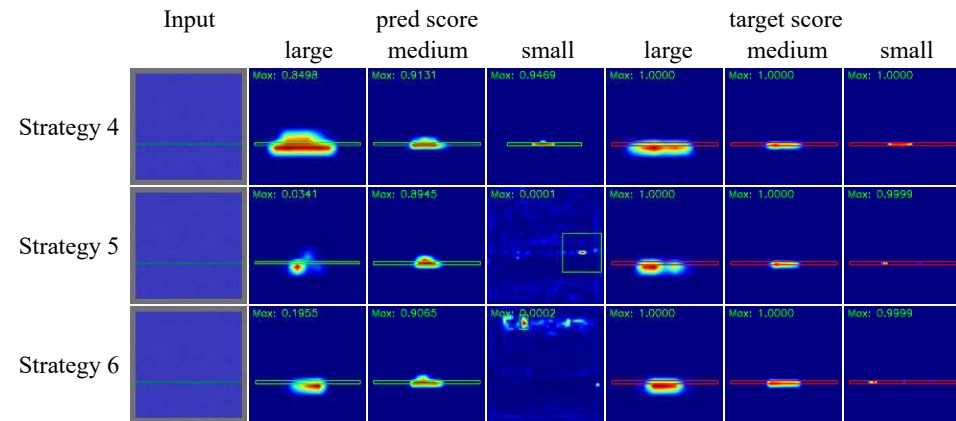

Figure 5: Visualize detectors' outputs with LWA at the last epoch in slim object.

Figure 5 shows the full time, 20% time, and 50% time LWA strategies, namely Strategy 4, Strategy 5 and Strategy 6 in Table 1. It shows the last epoch's heatmap rather than the 10th epoch in Figure4. We can seen that after appling LWA, all three detectors can be matched grids and the pred score of 20% time and 50% time LWA has the largest value at the medium detector, which is exactly what we expect.

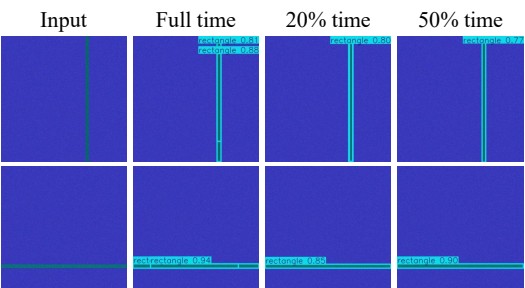

Figure 6: Detection results with different close time for LWA.

We also compare the prediction results of full time, 20% time and 50% time LWA. It can be clearly seen that there are multiple overlapping boxes in the predict result using full time LWA as is shown in Figure 6. This is because small detector also predicted these slim bboxes, and the confidence is not low. However, its predicted boxes are far from precise, resulting in a large number of duplicate boxes with low iou. While the prediction results of 20% and 50% are much cleaner.

### 3.3 INFLUENCE OF REGMAX

In addition, we also explore the influence of regmax. Is it that as long as regmax is set large enough, the performance can be comparative with our proposed method?

Yet, increasing regmax can indeed enhance the mAP of slim targets as is shown in Table 2 . However, when it is increased to 128, the mAP no longer increases and its highest mAP is still lower than ours(**0.995** mAP@50 **0.887** mAP50-95). Note that further increasing regmax is meaningless because regmax only affects the maximum width and height the pred bbox can represent and 128 is already sufficient to represent the biggest bbox. This indicates that simply increasing regmax cannot maximize the detection performance of slim objects.This is because slim objects detection needs larger reception field, small detector fails to detect them accurately even with large regmax.

Table 2: Influence of regmax

| regmax | mAP@50 | mAP@50-95 |
|--------|--------|-----------|
| 16 | 0.755 | 0.323 |
| 30 | 0.995 | 0.725 |
| 64 | 0.991 | 0.765 |
| 128 | 0.995 | 0.695 |

### 3.4 PREDICT RESULTS ON SYNTHETIC SLIM OBJECT DATASET

Table 3: mAP Values of Ours and Baseline in norm object Dataset.

| | mAP@50 | mAP@50-95 |
|--------|--------|-----------|
| Baseline | 0.995 | 0.889 |
| SODO (Ours) | 0.995 | 0.889 |

The detection examples, of our model and the baseline model, in both slim object and norm object datasets are shown in Figure 7. Table 3 shows the mAP of baseline and Ours in norm object dataset, for mAP of slim object dataset, refer to Table 1.

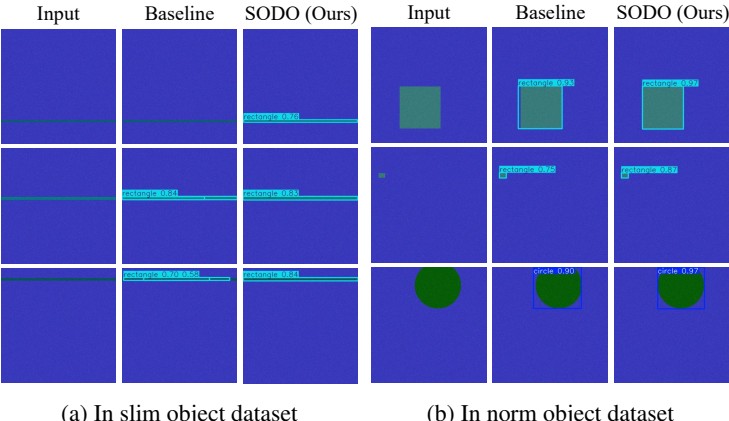

(a) In slim object dataset          (b) In norm object dataset

Figure 7: Detection examples for slim objects and norm objects

It can be seen from Figure 7(a) that the detection result of the baseline here contains a large number of low-quality detection bboxes, and even a slim object is not detected, while our detection results are very clean. Moreover, it can be seen from Figure 7(b) and Table 3 that our method does not affect the detection of common objects, and our detection results are basically the same as those of the baseline model.

### 3.5 EVALUATION ON PUBLIC DATASET

As is shown in Table 4, we presented the performance of our SODO method on the COCO dataset and the publicly available datasets from Roboflow. We compared the performance of our method on v11, v12, and v13, all of which demonstrated significant improvements in the detection of slim objects.

Figure 8 shows the predict result of COCOSlim dataset, it can be seen that our method significantly outperforms the baseline in detecting slim objects like train, and its confidence level is much higher

Table 4: Evaluation on public dataset

| | COCO val | | COCOSlim | | pole_detection test | | tans test | |
|---|---|---|---|---|---|---|---|---|
| | $mAP_{50}$ | $mAP_{50-95}$ | $mAP_{50}$ | $mAP_{50-95}$ | $mAP_{50}$ | $mAP_{50-95}$ | $mAP_{50}$ | $mAP_{50-95}$ |
| v11-L | 0.702 | 0.534 | 0.721 | 0.562 | 0.824 | 0.569 | 0.695 | 0.494 |
| v11-L (SODO) **Ours** | **0.705** | 0.530 | **0.731** | 0.561 | **0.836** | **0.580** | **0.712** | **0.501** |
| v12-L | 0.708 | 0.538 | 0.724 | 0.559 | 0.813 | 0.566 | 0.705 | 0.489 |
| v12-L (SODO) **Ours** | **0.711** | 0.537 | **0.731** | 0.558 | **0.832** | **0.578** | **0.729** | **0.499** |
| v13-L | 0.709 | 0.581 | 0.740 | 0.577 | 0.816 | 0.568 | 0.738 | 0.531 |
| v13-L (SODO) **Ours** | **0.712** | 0.580 | **0.746** | 0.577 | **0.825** | **0.574** | **0.742** | **0.532** |

than that of the baseline. Visual predict results of pole_detection and tans dataset are shown in Appendix B

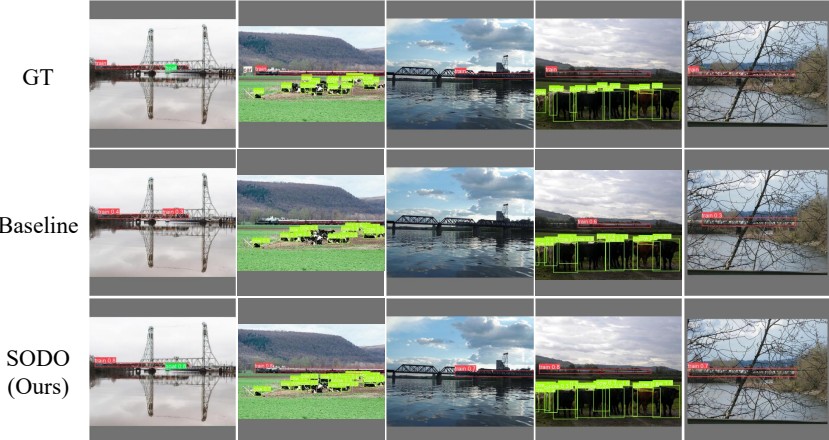

Figure 8: Detection results in COCOSlim.

## 4 CONLUSION AND LIMITATION

We propose SODO method to address the poor detection performance of YOLO with TOOD for slim objects. Firstly, through the construction of a dedicated SOA matching mechanism, slim objects can be effectively matched to grids in the deep detectors. This enables the model to better learn the characteristic features of slim objects. Moreover, Direct Offset & Center Offset are proposed to represent the pred bboxes. Finally, the LWA method is proposed, which is designed to normalize the target score for the three detectors independently. As a result, during the early stages of training, it enables medium and large detectors to learn relevant features effectively.

Our method can be extended to the Oriented bboxes (OBB) task. We will continue to work on this part in the future. Moreover, compared with the baseline, our method is more likely to cause the NAN loss problem, so we have to disable AMP training for now. We will conduct deeper research on the root cause of this problem in the follow-up work.

**Reproducibility Statement:** The code for this research is open-source and can be found at https://anonymous.4open.science/r/SODO-E6B5.

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

## A    RELATE WORKS

### A.1    YOLO FRAMEWORK

As shown in Figure 9, the YOLO models are typically composed of three core parts: Backbone, Neck, and Head. The Backbone is responsible for feature extraction. The Neck performs multi-scale feature fusion. The Head is responsible for converting the features extracted from the previous parts into specific detection information and supports multiple tasks such as detection, segmentation, and pose estimation. The Head consists of three scale detectors: large, medium, and small, each tailored to detect objects of varying sizes. Specifically, the large detector has a feature map of relatively small sizes, yet it is designed to detect large objects. In contrast, the small detector has a feature map of relatively large size, and its function is to detect small objects.

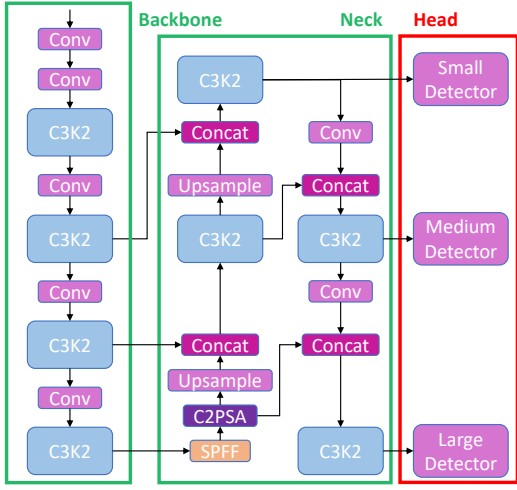

Figure 9: YOLO v11 framework.

### A.2    TOOD

TOOD (Feng et al., 2021) aligns the classification and localization through the Task-aligned Head (T-head) and Task Alignment Learning (TAL), so that the bbox output by the grid with the highest confidence is also the most precise.

### A.2.1 T-HEAD

**Task-interactive features extraction:** Multi-scale interaction features are extracted from FPN features through multi-layer convolutions (such as six layers of 3×3 convolutions) to enhance the shared information between tasks.

**Task-aligned Predictor (TAP):** The layer attention mechanism is used to dynamically allocate task-specific features, generate classification and localization predictions respectively, and reduce feature conflicts.

### A.2.2 TAL

**Alignment metric:** As is shown in the Equation 10, by combining the classification score and the Intersection over Union (IoU) value, define the alignment metric $t$ to dynamically select high-quality anchors, and use the normalized $t$ as the label for the score as is shown in Equation 11 .

$$t = s^\alpha \times u^\beta \tag{10}$$

$$t_{norm} = \frac{t}{max_{s,m,l}(t)} \tag{11}$$

where $s$ and $u$ denote a classification score and an IoU value, respectively. $\alpha$ and $\beta$ are used to control the impact of the two tasks in the anchor alignment metric. s, m, l denote the small, medium, large detectors respectively, $max_*(t)$ denotes the maximum value of t among detector *. Notably, $t$ plays a critical role in the joint optimization of the two tasks towards the goal of task alignment. It encourages the networks to dynamically focus on high-quality (i.e., task-aligned) anchors from the perspective of joint optimization.

**Sample allocation:** A target can be matched to a grid only when target's bbox pass through the grid center, as is shown in Figure 2(a).

**Loss function:**

Classification loss: Replace the label with $t$ as the target score to supervise the learning of the predict score.

Regression loss: Weight the GIoU Loss (Rezatofighi et al., 2019) with $t$ to improve the positioning accuracy of the aligned anchors.

### A.3 DISTRIBUTION FOCAL LOSS (DFL)

As is illustrated in Equation 3 and Figure 10, DFL (Li et al., 2023) is mainly applied in the bbox regression process of object detection. The main function of the DFL is to correct the errors of the model when predicting the bboxes of objects.

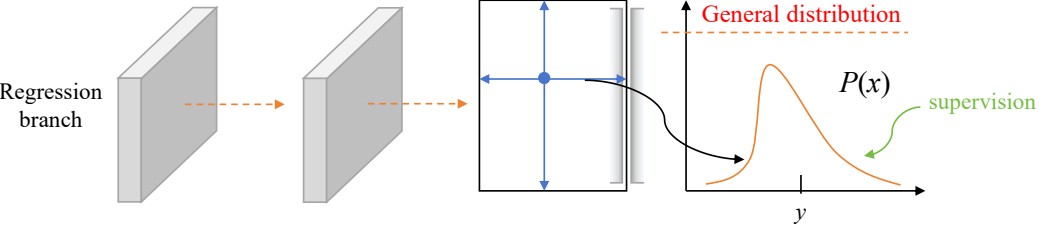

Figure 10: DFL Loss.

# B    ADDITIONAL EXPERIMENTAL RESULTS

In addition to the results presented in the paper, we show supplement results for the pole_detection and tans datasets in Figure 11 and Figure 12. It can be seen from pole_detection detection results, ours detection results are much cleaner than baseline and get higher confidence, and tans detection results show that ours detection results are much better than baseline in slim objects.

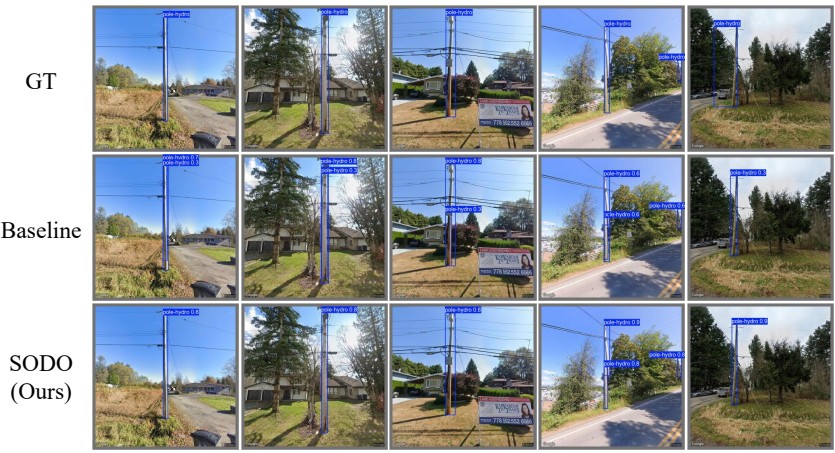

Figure 11: Detection results in pole_detection.

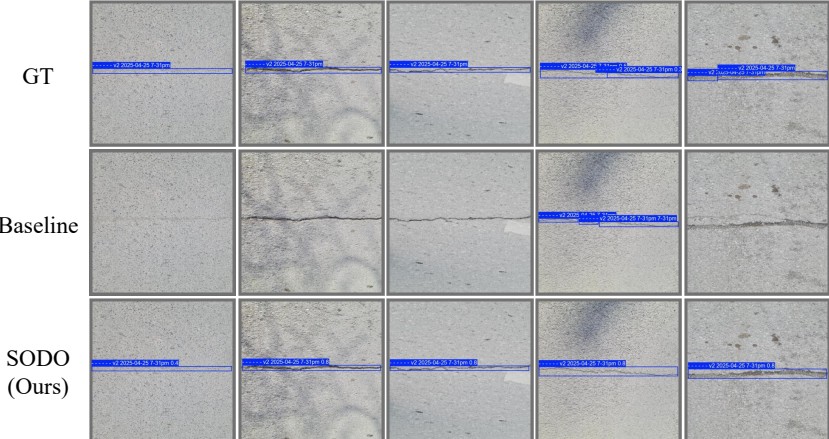

Figure 12: Detection results in tans.

# C    THE USE OF LANGUAGE MODELS (LLMS)

During the writing of this paper, to optimize the English expression, LLMs (WeTab AI, DeepSeek, Hunyuan) were used to translate some Chinese content and conduct grammar checks and corrections on the English text. However, all experimental designs, data collection and analysis, as well as conclusion derivations, were independently completed by the authors.

