# OpenReview forum: "Slim Object Direct Offset (SODO): Why YOLO with TOOD struggles to detect slim objects? A new matching approach for slim objects"
_ICLR.cc/2026/Conference — ICLR 2026 Conference Withdrawn Submission_

### Official Review · Reviewer_ywSV · 2025-10-20

**Soundness:** 1
**Presentation:** 1
**Contribution:** 1
**Rating:** 2
**Confidence:** 4

**Summary:**

The paper presents a label assignment method called slim object assignment (SOA), which is built on TOOD to improve the detection performance for slim objects. A anchor point is determined as a positive if the slim object passes through the corresponding area within the point. The author also design a layer wise assignment (LWA) performs normalization independently for small, medium, and large detectors at the beginning of training. The experiments are conducted on a synthetic dataset and three realistic datasets.

**Strengths:**

The study of improving slim object detection is meaningful.

**Weaknesses:**

1. The novelty is limited. The label assignment is an improved version of TOOD only. By the definition, it is a static label assignment algorithm, not a dynamic one. The paper does not verifies the superiority of the proposed method over the dynamic algorithms such as OTA [r1], DW [r2].

2. The experiments and the analyses are mainly conducted on a toy, unrealistic, and synthetic dataset, making the soundness of the method less convincing. Additionally, the experimental results on COCO datasets shows a degradation on AP. Even to the slim objects, AP is not improved.

3. Important comparison is missing. The paper does not compare the proposed method with the dynamic label assignment, e.g., OTA, DW.

4. The reported results is wrong. YOLOv13-L cannot achieves an AP of 58.1 both in the paper and in its official repository.

5. The writing of this paper is very rough and is filled with colloquial expressions. Many expressions are not formal and even ambiguous. This makes the paper more like a technical blog rather than a scientific paper. For example, it would be better call YOLOv11, YOLOv12, YOLOv13 instead of v11, v12, v13. Mathematical symbols are inconsistent, e.g., it is i-th probability in the text but $i$ in the equations. There is an error in Eqn. 1, specifically in the term "$y_{rd}$".

[r1] Ge Z, Liu S, Li Z, et al. Ota: Optimal transport assignment for object detection. CVPR 2021.

[r2] Li S, He C, Li R, et al. A dual weighting label assignment scheme for object detection. CVPR 2022.

**Questions:**

see weaknesses above

---

### Official Review · Reviewer_b6jM · 2025-10-27

**Soundness:** 2
**Presentation:** 1
**Contribution:** 1
**Rating:** 2
**Confidence:** 4

**Summary:**

This paper aims to improve the detection performance of slim objects based on current YOLO framework. This article points out that the existing framework has design flaws when detecting slim objects. It also proposes its own solution to the flaws.

**Strengths:**

Exploring the design flaws of existing detection frameworks is an interesting but also challenging problem. This article proposes a feasible solution to the blind spots of common detection frameworks.

**Weaknesses:**

1. The problem being solved is too narrow.
2. The experimental results in Table 4 do not show significant improvement.
3. Writing is a bit bad. The spelling in formula 1 is wrong. It is x_rb or x_rd?

**Questions:**

1. Why we need to study this problem. What percentage of slim objects are detected in general detection? What scenarios and categories are they usually in? If we abstract it to slim object detection, does it include both curved and straight objects?
3. Figure 2 primarily illustrates the advantages of the proposed method over existing methods when the target object and anchor mismatch occurs. However, as the authors previously described, a crucial characteristic of slim objects is their aspect ratio. So, what advantages does the proposed method offer over existing methods when the aspect ratios differ significantly but the anchors match?
4. Can the proposed method be extended to a general method for irregular and uncommon objects?

---

### Official Review · Reviewer_BxFr · 2025-10-30

**Soundness:** 2
**Presentation:** 2
**Contribution:** 2
**Rating:** 4
**Confidence:** 4

**Summary:**

The paper proposes SODO, a method to address detection issues in slim objects (AR > 20) for TOOD-based YOLO models. Specifically, the method introduces several novel ideas: Firstly, SOA (Slim Object Assignment) changes the positive sample assignment criterion from “center-passing” to “overlap with the grid cell,” enabling slim objects to receive positive samples across multiple scales. Secondly, new bbox parameterizations, Direct Offset, allow negative offsets, and Center Offset ensures non-negative shifts, although the latter converges more slowly. The method also introduces LWA (Layer-Wise Assignment), which normalizes alignment scores independently per head in the early training stages, preventing small scales from dominating the scoring and impeding medium/large scale learning. Finally, Remove Overlaps eliminates the extra IoU multiplication in the final target score to improve early training supervision. Experimental results show slight mAP@50 gains (+0.3%) on COCO and more significant improvements on slim-object subsets and various industry-specific datasets.

**Strengths:**

* The paper clearly identifies the shortcomings of TOOD-based models in detecting slim objects and provides a detailed analysis of the root causes.
* The design of SODO is highly implementable. The definition of SOA is clear and can be directly integrated into existing YOLO codebases, and the incorporation of Direct/Center Offset naturally fits within the existing DFL framework, making it easy for practical adoption. The small change of removing IoU multiplication also significantly improves the early training process, enhancing overall stability.
* The paper provides a detailed discussion of how LWA addresses head imbalance during early training. By independently normalizing the alignment scores for different scales during the first 20% of epochs, LWA effectively prevents small scales from dominating and allows medium/large scales to catch up. This approach is supported with solid reasoning and is practical for improving training dynamics.
* The paper conducts ablation studies to assess the effects of various strategy combinations and visualizes how each configuration impacts detection performance.
* The paper evaluates the method across multiple versions of YOLO and reports results on COCO, COCO-Slim, and public datasets, demonstrating SODO’s effectiveness in slim-object detection.

**Weaknesses:**

* Modest and sometimes negative changes on COCO mAP@50–95. Improvements concentrate on mAP@50 and slim subsets, with no multi-seed variance reported to assess significance.
* The paper also mentions that mixed-precision training is prone to NaN/Inf errors, and while a temporary solution (disabling AMP) is provided, this approach increases memory usage and training time significantly. The paper does not compare other approaches (such as gradient clipping) in terms of training stability and efficiency.
* Regarding the dataset, the paper primarily uses a synthetic slim-object dataset, where each image contains only one object, and the aspect ratio is set to 80. This setup may not generalize well to multi-object or occluded scenarios. The lack of validation in more complex scenes (e.g., multi-object, occlusion, dense scenes) may result in a model that underperforms in real-world applications.

**Questions:**

* Under SOA, could the author provide the average and maximum number of positives per GT at each scale and how these evolve during training? If there is class imbalance, has the author tried top-$k$ or dynamic $K$ to control the number of positive samples?
* Does the assumption of a lower bound for negative offsets $(-0.5)$ depend on stride or coordinate normalization? Does this assumption hold at different resolutions?
* Regarding the geometric consistency of Direct Offset, how does the author ensure the correct order of $\mathrm{tl}/\mathrm{rb}$ during decoding?
* Could the author provide a deeper theoretical analysis for the Remove Overlaps method? What is the interaction between this method and $\alpha,\beta$, and how does it affect the learning curve? Are there ablation studies on the coupling with target alignment scores?
* When using mixed-precision training, is disabling AMP the only solution? Has the author tried other methods (such as gradient clipping) to improve stability?
* Could the author provide quantified inference-time data? Do changes in SOA and LWA increase proposal density or cross-scale duplicates? Can the author quantify inference speed (FPS), memory usage, NMS time, and deduplication strategies in the experiments?
* Since the improvements on COCO are small, can the author provide variance across multiple runs (at least 3 random seeds)? If there is significant variation, how does the author ensure the stability of the results?
* Given that the synthetic dataset contains only one object per image with extreme AR, does the author have validation results on multi-object, occluded, or dense scenes? How does the author demonstrate SOA’s effectiveness in these scenarios, especially regarding positive sample selection and false positive issues?

---

### Official Review · Reviewer_3FnX · 2025-11-01

**Soundness:** 2
**Presentation:** 1
**Contribution:** 1
**Rating:** 2
**Confidence:** 4

**Summary:**

This paper analyzes why recent YOLO models (v8–v13) that rely on the TOOD (Task-Aligned One-Stage Object Detection) mechanism perform poorly on slim objects with extreme aspect ratios (>20). The authors attribute this to TOOD’s grid-bbox matching rule, which often fails to assign such objects to any grid. They propose Slim Object Direct Offset (SODO), incorporating:
(1) Slim Object Assignment (SOA): a relaxed matching rule allowing any grid intersecting the object;
(2) Direct Offset and Center Offset: new bounding-box representations allowing negative offsets;
(3) Layer-Wise Assignment (LWA): independent normalization for small, medium, and large detectors early in training; and
(4) Overlap removal: intended to improve convergence.

Experiments on COCO, a filtered COCOSlim subset, two Roboflow datasets, and synthetic data show modest improvements in AP@50 for slim objects while maintaining comparable accuracy on normal objects.

**Strengths:**

* The paper clearly identifies a niche but real weakness in the TOOD matching mechanism.
* The proposed modifications are simple, easy to implement, and empirically improve detection of extremely elongated objects.
* Ablation studies and visualizations seem comprehensive.

**Weaknesses:**

Incremental and heuristic contribution: The proposed changes, namely SOA, LWA, and modified offsets, are largely ad hoc tweaks to existing matching heuristics. They do not introduce new learning principles, architectures, or theoretical insights.

Marginal empirical gains: The reported improvements on COCO and COCOSlim (+0.3% to +1% mAP@50) are minor and maybe within typical training randomness/noise; statistical significance is not demonstrated.

Limited scope and generality: The work specifically targets “slim” object detection within YOLO+TOOD, with little evidence that the method generalizes to other detectors or object geometries.

Weak relevance to ICLR: The contribution lies in applied object detection engineering rather than advancing core ICLR topics, such as representation learning or optimization. THe paper is more suitable for an applications conference such as WACV.

Implementation fragility: The need to disable mixed-precision training to avoid NaN losses suggests numerical instability and limits practicality.

Writing quality and clarity: While the structure is clear, language issues and superficial mathematical depth detract from the paper’s technical rigor.

**Questions:**

I do not have any specific questions but if the authors want, they can address the issues I raised above.

---

### Note · Authors · 2025-11-14

I have read and agree with the venue's withdrawal policy on behalf of myself and my co-authors.